



**Molecular insight into aqueous-phase photolysis and photooxidation**
**of water-soluble organic matter emitted from biomass burning and**
**coal combustion**
Tao Cao[1], Cuncun Xu[1,2], Hao Chen[1,2], Jianzhong Song[1,3,*], Jun Li[1,3], Haiyan Song[4],
Bin Jiang[1,3], Yin Zhong[1,3], Ping'an Peng[1,2,3]
[1]State Key Laboratory of Advanced Environmental Technology and Guangdong
Provincial Key Laboratory of Environmental Protection and Resources Utilization,
Guangzhou Institute of Geochemistry, Chinese Academy of Sciences, Guangzhou
510640, China
[2]University of Chinese Academy of Sciences, Beijing 100049, China
[3]Guangdong-Hong Kong-Macao Joint Laboratory for Environmental Pollution and
Control, Guangzhou 510640, China
[4]School of Chemistry, South China Normal University, Universities Town,
Guangzhou 510006, China
[*]*Correspondence to:* Jianzhong Song, E-mail: songjzh@gig.ac.cn.





**Abstract**
Biomass and coal combustion represent substantial contributors to atmospheric
water-soluble organic matter (WSOM). It experienced intense photochemical
oxidation once entered atmospheric environment, but the resulting changes in WSOM
are largely unclear. This study examines the changes in the optical properties,
fluorophores, and molecular composition of WSOM derived from the combustion of
biomass (specifically rice straw, RS) and coal (from Yulin, YL) during aqueous
photolysis and hydroxyl radical ($OH$) photooxidation. The results indicate that
photochemical aging induces distinct changes in the light-absorbing properties of RS
and YL WSOM, characterized by pronounced photobleaching in RS WSOM and
photoenhancement in YL WSOM. Additionally, more pronounced alterations were
observed during $OH$ photooxidation than direct photolysis, for both RS and YL
WSOM. Furthermore, a greater proportion of molecules in both RS (61.6%) and YL
(65.0%) WSOM were degraded during $OH$ photooxidation compared to photolysis
(14.9% and 23.1%, respectively), resulting in products with larger molecular weight
and higher oxidation levels, including tannin-like substances and newly formed black
carbon-like compounds, whereas the products of photolysis were characterized by
relative minor alteration. These findings provide new insights into the photochemical
evolution of combustion-derived WSOM and help to predict its effects in
environmental and climate changes.



**1. Introduction**

Water-soluble organic matter (WSOM) consists of diverse array of polar organic
species, which is ubiquitous in atmospheric aerosols, cloud, fog, and rain waters (Sun
et al., 2023; Wang et al., 2019). WSOM can not only alter the hygroscopicity and
surface tension of aerosol, influence the formation of cloud condensation nuclei, but
also has significant effects on the radiative forcing of aerosols, thereby playing crucial
roles in atmospheric environment and climate change (Sun et al., 2011; Chen et al.,
2019; Lee et al., 2022). Due to its high reactivity, WSOM also contributes to
atmospheric chemistry and the formation of organic aerosols. Moreover, WSOM has
the potential to catalyze the generation of reactive oxygen species, posing adverse
impacts on human health (Bhattu et al., 2024; Bates et al., 2019).
Multiple sources of WSOM have been identified, including primary emissions
from biomass burning (BB), coal combustion (CC), vehicular emissions, and
secondary formation through the photochemical transformation of volatile organic
compounds (Tang et al., 2020; Jiang et al., 2023; Cao et al., 2023). Among these
sources, BB has been recognized as a significant contributor to atmospheric WSOM
in numerous regions, including East Asia, Southeastern Asia (Liu et al., 2021; Zheng
et al., 2017), the Amazon rainforest (Malavelle et al., 2019), and North America
(Gallo et al., 2023; Ceamanos et al., 2023). Furthermore, domestic coal combustion
also serves as a crucial primary source of atmospheric WSOM in northern China and
India (Bikkina et al., 2020; Liu et al., 2022), as well as in Poland (Casotto et al., 2023).
It is important to note that the combustion-derived primary WSOM experiences





considerable aging upon entering the atmosphere (Sumlin et al., 2017; Schnitzler et al.,
2022). For instance, studies have reported a marked decrease in the light absorption of
water-soluble brown carbon (BrC) during transport over distances exceeding 6000 km
from the Indo-Gangetic Plain to the Himalayan region (Dasari et al., 2019; Choudhary
et al., 2022). Additionally, observations of wildfire plumes in North America have
demonstrated a reduction in the mass absorption coefficient as the plume ages (Bali et
al., 2024). Nonetheless, the concentrations, light absorption properties, and chemical
characteristics of WSOM undergo significant alterations throughout the atmospheric
aging process.
Field and laboratory studies demonstrated that aqueous photochemical processes
including direct photolysis and secondary photochemistry involving with oxidants
(e.g., hydroxyl radical (·OH), $O_3$), are ubiquitous and play a significant role in the
transformation of atmospheric WSOM (Hems et al., 2021; Manfrin et al., 2019).
Research conducted by Cai et al. (2020) revealed that the aqueous photochemistry of
BB WSOM can produce highly oxygenated compounds, which subsequently enhance
the oxidation state of WSOM in atmospheric samples. Furthermore, the ·OH
photooxidation of BB-derived organic species (e.g., 4-methylsyringol, eugenol) has
been found to form light-absorbing products, indicating a potential pathway for
secondary organic aerosol (SOA) (Liu et al., 2022; Li et al., 2023; Arciva et al., 2022).
Additionally, the ·OH photooxidation of freshly emitted BB WSOM initially result in
an increase in its absorption capacity, which is later followed by a photobleaching
process during the photoaging (Hems et al., 2021; Wong et al., 2017). These finding



underscore the dynamic nature of WSOM due to the photolytic aging, however,
further insights into the molecular transformations leading to these observations
remain unclear. Moreover, while the chemical composition of WSOM emitted from
BB and CC differs, it remains unclear whether distinct classes of molecules exhibit
varying behaviors during photochemical processes.
To address these inquires, the photochemical aging of WSOM emitted from both
biomass burning and coal combustion was systematically investigated through direct
photolysis and OH photooxidation in the aqueous phase. The objectives are (1) to
compare the optical evolution of BB and CC WSOM under the photolysis and  OH
photooxidation; and (2) to elucidate photochemical transformation of BB and CC
WSOM at a molecular level by using fourier transform ion cyclotron resonance mass
spectrometry (FT-ICR MS). The information obtained will enhance understanding of
the atmospheric oxidation processes of combustion-derived WSOM and their
subsequent environmental and climatic effects.

**2. Materials and methods**
**2.1. Preparation of WSOM samples**
Rice straw (RS) and Yulin coal (YL) were selected as representative biomass and
coal fuel materials for the preparation of combustion-derived WSOM samples. These
materials are commonly utilized for heating and cooking in rural households,
particularly during winter season in northern China. Additionally, RS residue is also
burned in agriculture field (Zhang et al., 2023; Huang et al., 2022). The smoke



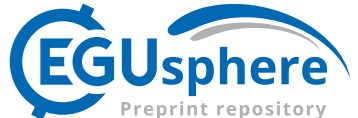

samples emitted from combustion process were collected in a laboratory-controlled
combustion system in our laboratory and more detailed information can be seen in our
previous studies (Cao et al., 2021; Li et al., 2018). Immediately after collection, the
filters were wrapped with baked aluminum foil and stored in a refrigerator (−20 ℃).
Prior to conducting photolysis and photooxidation experiments, the WSOM
fraction was extracted using ultrapure-water. Briefly, filter sample was cut into pieces
and placed in 100 mL glass bottle, to which 60 mL ultrapure water was added. After
ultrasonically extracted for 30 min, the extract was filtered through a 0.22 μm
polytetrafluoroethylene syringe filter (Anpel, ANPEL Laboratory Technology
(Shanghai) Inc.). The organic carbon concentration of WSOM solution was measured
before photochemical reaction and diluted to 20 mgC/L by ultrapure water, in
accordance with preliminary experimental protocols (Gu et al., 2024; Zhang et al.,

119   2022).

**2.2. Photolysis and OH photooxidation experiment**
The photolysis and OH photooxidation experiments were conducted in a
photoreactor, where quartz cell containing WSOM solution were continuously
exposed to radiation. Briefly, 100 mL of 20 mgC/L WSOM solution was magnetically
stirred in a 250 mL cylindrical quartz cell equipped with water circulating jacket to
maintain a constant temperature of 25 ℃. For the photolysis experiment, the WSOM
solution was irradiated from the top by Xenon lamp (PL-XQ500W, Beijing Princess
Technology co. ltd) with an output energy of 500W at 0.2 m. The irradiation energy at
the water surface is 12.5 mW/cm$^2$ in the range of 290−400 nm. For the ·OH



photooxidation experiments, 3mM $H_2O_2$ was added to the WSOM solution as a
photolytic source of OH radicals upon irradiation (Zhao et al., 2015; Arciva et al.,
2024).For each experiment, 4 mL samples were withdrawed periodically (0, 1, 2, 4, 8,
12, 24h) from the reactor and then diluted to 20 mL for further analysis. The
photolysis and OH photooxidation experiment were both carry out the dark control
synchronously follow the conditions as introduced above. The results showed that no
significant changes were observed for the organic carbon content and the UV-vis
absorption of WSOM within the reaction time.
**2.3. Spectroscopy measurement**
The UV-vis absorption of WSOM was measured using a UV-vis
spectrophotometer (UV-2600i, Shimadzu, Japan) within the wavelength range of 200
− 700 nm. Excitation-emission matrix (EEM) spectra were recorded by
three-dimensional fluorescence spectrophotometer (Aqualog, HORIBA Scientific,
USA). The scanning ranges for excitation (Ex) and emission (Em) were 240–800 nm
and 250-800 nm, respectively. Ultrapure water ($18.2M\Omega cm^{-1}$) was used as blank
reference and correcting the fluorescence intensity unit by the Raman peak area of
water (R.U.). In addition, the corresponding absorption spectra were used to correct
the EEM for inner-filter effects (IFEs) according to the previous studies if the
absorbance was higher than 0.05 at 250 nm (Tang et al., 2020; He and Hur, 2015;
Murphy et al., 2013). The PARAFAC modeling procedure was conducted in
MATLAB 2021b (Mathwork.Inc, USA) by the drEEM toolkit (Murphy et al., 2018;
Pucher et al., 2019). More information and data processing details are provided in



Text S1 of Supporting information (SI).

**2.4. High-resolution mass spectrometry analysis**

The molecular characteristics of WSOM before and after photolysis and ·OH
photooxidation were measured with a solariX XR FT-ICR MS (Bruker Daltonik
GmbH, Bremen, Germany) equipped with a 9.4T refrigerated actively shielded
superconducting magnet and a Paracell analyzer cell. The WSOM samples used for
FT-ICR MS analysis were desalted by solid phase extraction cartridge (Oasis HLB,
200 mg, Waters, Milford, MA, USA) as introduced in our previous studies (Song et al.,
2019; Song et al., 2018; Song et al., 2022). The detailed measurement condition and
the calculation of corresponding indexes (e.g. double bond equivalents (DBE) and
modified aromaticity index ($AI_{mod}$)) are described in Text S2 in SI. For better
elucidate the transformation of RS and YL WSOM, the photochemical resistant,
degraded, and produced molecules were investigated (Fan et al., 2024; Gu et al.,

2024).


**3. Results and discussion**

**3.1. Effect of photolysis and ·OH photooxidation on the light absorption of**

**WSOM**

The absorption spectra of RS and YL WSOM during photolysis and ·OH
photooxidation are illustrated in Figure 1a-d. It can be observed that the absorbance of
RS WSOM gradually decrease as aging time increasing during both photolysis and
·OH photooxidation, indicating substantial photobleaching (Fan et al., 2024; Zhao et





al., 2022). Moreover, the reduction in absorbance during ·OH photooxidation is more
obvious than that in photolysis, indicated that RS WSOM undergoes greater
degradation during ·OH photooxidation. In contrast, the absorbance of YL WSOM
present different variation during the photolysis and ·OH photooxidation. Specifically,
the absorbance in the short wavelength range of 210-240 nm decreases gradually with
aging time, while the absorbances at wavelengths exceeding 360 nm increase. This
phenomenon is characteristic of photoenhancement, which aligns with finding
reported in previous studies concerning nitrate-mediated photooxidation of guaiacol
and 5-nitroguaiacol as well as photooxidation of mixed aromatic carbonyls (Go et al.,
2024; Yang et al., 2021).

To quantitatively assess the changes in light-absorbing substances during

photolysis and ·OH photooxidation, the absorption coefficients at 254 nm ($\alpha_{254}$) and
365 nm ($\alpha_{365}$) were calculated (Fan et al., 2024; Zou et al., 2023). As shown in Figure
1e, the $\alpha_{254}$ values for both RS and YL WSOM consistently decline during photolysis
and  OH photooxidation, with a more significant reduction observed during ·OH
photooxidation. These results are consistent with earlier studies on  OH oxidation and
photochemical oxidation of BB WSOM, indicating that the presence of ·OH radicals
accelerate the degradation of aromatic structures within WSOM (Fan et al., 2024; Ye
et al., 2020). Additionally, the reduction of $\alpha_{254}$ values was always greater for RS
WSOM than for YL WSOM, suggesting that RS WSOM are more susceptible to
photochemical degradation.

The $\alpha_{365}$ values for RS and YL WSOM exhibit different variations under



photolysis and ·OH photooxidation. As illustrated in Figure 1f, the $\alpha_{365}$ value for RS
WSOM gradually decrease with prolonged photolysis, while it initially increased
slightly before decreasing during ·OH photooxidation. Similar observation has been
made in the photochemical aging of wood smoke BrC and monomeric phenolic
compounds, suggesting the formation of new compounds with significant
light-absorbing capacity during the initial stage of ·OH photochemical reaction (Hems
et al., 2021; Wong et al., 2017; Lee et al., 2014). In contrast, the $\alpha_{365}$ values of YL
WSOM present markedly different trends, increasing during 24h photolysis and
initially rising for 12h before decreasing from 12h to 24h during ·OH photooxidation.
These results indicates that the products generated from photochemical reaction of YL
WSOM possess enhanced light absorbance in the near-UV and visible regions, which
are also observed in the aqueous phase oxidation of aromatic compounds such as
phenols (Arciva et al., 2024; Smith et al., 2016). The proposed mechanism may
involve the aromatization of phenolic compounds and ·OH-functionalization of
aromatic compounds, leading to the formation of the strong light-absorbing
substances at longer wavelength (Li et al., 2023).
**3.2. EEM-PARAFAC of WSOM during the photolysis and ·OH photooxidation**
The EEM-PARAFAC model has successfully identified three distinct fluorescent
components (C1–C3) within RS and YL WSOM. As shown in Figure 2a, C1 displays
excitation/emission peaks at Ex/Em = 270/325 nm, which are attributed to protein-like
substances, including tyrosine-like substances (Podgorski et al., 2018; Hu et al., 2023),
as well as non-nitrogenous containing species such as phenol-like compounds (Cao et



al., 2023). C2 (240, 320/420 nm) and C3 (240/350 nm) both assigned to humic-like
substances (Hu et al., 2023; He et al., 2023; Fan et al., 2021). Due to the fluorescence
distributed at longer wavelengths are mainly associated of larger molecular weight
and highly oxygenated of fluorophores (Cao et al., 2023), thereby suggesting that the
longer emission wavelengths of C2 might be associated with highly oxygenated
humic-like fluorophores with higher molecular weight and aromaticity, while C3
could be more relevant to less oxygenated structures and conjugated systems.
Furthermore, fluorophores contain same position with C2 have been observed during
the photooxidation of vanillic acid and ozone oxidation of BB BrC (Fan et al., 2021;
Tang et al., 2020), indicating C2 may be closely related with the products formed
through atmospheric oxidation processes.
To quantitatively access the changes in the distribution of fluorophores during the
photochemical process, total fluorescence intensity (TFI) was calculated. As depicted
in Figure 2d, the TFI values for RS and YL WSOM showed a comparable decline
during both photolysis and ·OH photooxidation, with a pronounced reduction during
·OH photooxidation. These results indicate that fluorophores are more susceptible to
degradation or quenching by ·OH attacks than by direct photolysis in both BB and CC
WSOM. On the one hand, more aromatic structures in WSOM may be disrupted by
·OH radical, resulting a more significant reduction in fluorophores. On the other hand,
the ·OH photooxidation also lead to an increase in carboxyl groups, which are the
typical electron-withdrawing groups. thereby contributing to a reduction or quenching
of fluorescence in WSOM.





Moreover, the relative contribution of the three fluorophores in RS and YL
WSOM varied throughout photochemical processes, with more significant changes
noted during ·OH photooxidation (Figure 2e-h). It is obvious that the increases in C2
are 49% and 56% for RS and YL WSOM during ·OH photooxidation, which is
significantly higher than that 5% and 14% during photolysis. These can be explained
by the formation of more highly oxygenated humic-like fluorophores due to ·OH
photooxidation (Zhang et al., 2022; Fan et al., 2024). In contrast, fluorophore C3
greatly declined by 35% and 56% for RS and YL WSOM, respectively, during ·OH
photooxidation. Previous studies have linked fluorophore C3 to less-oxygenated
fluorescent substances resulting from primary combustion (Cao et al., 2023; Chen et
al., 2016), which can be oxidized and gradually removed during the ·OH
photooxidation process. Comparatively, the contributions of three fluorescent
components in RS and YL WSOM both display relatively minor variations under
photolysis, suggesting the lower selectivity of photolysis. These notable variations in
both the subgroup and intensity of fluorophores suggest their potential utility as
indicator of the atmospheric oxidation processes experienced by fresh emissions (Fan
et al., 2024; Ye et al., 2025).
**3.3. Changes in molecular characteristics of RS and YL WSOM**
Figure 3 showed the FT-ICR MS spectra of RS and YL WSOM before and after
undergoing photochemical oxidation. A total of 5114 to 6383 molecules were
identified within the m/z range of 100-600, with a predominant concentration of peaks
observed between 150 to 400. These finding are indicative of the molecular



characteristics typical of organic compounds resulting from BB and coal combustion
emissions (Tang et al., 2020; Song et al., 2018; Song et al., 2022). The identified
formulae were categorized based on their elemental compositions into four groups:
CHO, CHON, CHOS, and CHONS (Tang et al., 2020; Song et al., 2018). As shown in
Figure 3, CHO and CHON compounds are the dominant compounds (95.8%-98.4%)
in RS WSOM, with minor fluctuations following photolysis and OH photooxidation.
These observations align with findings related to BB WSOM subjected to dark OH
oxidation and BB smoke aerosols in an oxidation flow reactor (Fan et al., 2024; Zhao
et al., 2022). In contrast to RS WSOM, YL WSOM contain not only high content of
CHO (47.6%) and CHON (33.1%), but also significant S-containing substances
(CHOS and CHONS, 19.2%). Figure 3 reveals notable differences in the composition
of compound groups within YL WSOM. The CHO compounds in fresh YL WSOM
are 47.6%, which increased to 76.1% and 84.2% after photolysis and OH
photooxidation, respectively. Whereas, the CHON compounds decreased from 33.1%
to 13.4% and 13.6%, respectively. Additionally, S-containing compounds
demonstrated a marked decrease following photolysis and OH photooxidation for YL
WSOM. These discrepancies may be attributed to the inherent differences in
molecular composition between RS WSOM and YL WSOM, which exhibit varying
sensitivities to photolysis and OH photooxidation.
**3.3.1. Molecular properties**
The intensity weighted average values of various molecular parameters,
including molecular weight ($MW_w$), elemental ratios, double bond equivalents





(DBE$_w$), modified aromaticity index (AI$_{mod, w}$) and oxidation state of carbon (OS$_{c, w}$)
of RS and YL WSOM before and after photochemical aging were summarized in
Table S1. It is evident that the molecular characteristics of WSOM underwent
significant alterations following photolysis and ·OH photooxidation. Specifically, the
MW$_w$ value of fresh RS WSOM is 252, which increased to 288 and 319 after
photolysis and ·OH photooxidation, respectively. The similar trend was observed for
YL WSOM, where the MW$_w$ values increased from 231 to 268 and 303, respectively.
These results align with previous studies indicating that the MW values of BB WSOC
increased after dark ·OH oxidation and photolysis (Fan et al., 2024; Wong et al., 2019).
Such changes may be attributed to the newly formation of higher MW molecules
through the oligomerization reactions and the resistance of high MW ones during
photochemical aging (Gu et al., 2024; Fan et al., 2024; Go et al., 2024; Waggoner et
al., 2015; Carena et al., 2023). Furthermore, it is noteworthy that the MW$_w$ values for
both RS and YL WSOM following ·OH photooxidation were greater than that after
photolysis, suggesting that ·OH photooxidation exerts a more pronounced aging
effect.
As detailed in Table S1, the AI$_w$ value of fresh RS WSOM is 0.44, which
subsequently decreased to 0.42 and 0.36 after photolysis and OH photooxidation,
respectively. Similar variation was noted for YL WSOM, where the AI$_w$ value
decreased from 0.56 to 0.52 and 0.47, respectively. Moreover, the reduction in AI$_{mod,w}$
values were more pronounced for both RS and YL WSOM subjected to OH
photooxidation. These results indicate that the aromatic structures within WSOM



were disrupted during photochemical processes, with OH photooxidation resulting in
more significant breakdown (Zhao et al., 2022).
The $O/C_w$ and $OS_{c, w}$ values were used to estimate the oxidation degree of the
formulae in WSOM. As shown in Table S1, the $O/C_w$ of RS WSOM increased from
0.38 to 0.43 and 0.59 after photolysis and OH photooxidation, respectively,
indicating an increase in the number of O atom within the molecular post-oxidation.
Notably, the $OS_{c,w}$ values exhibited a similar trend to that of $O/C_w$. These observations
are consistent with findings related to BB WSOC under dark oxidation and the
photochemical transformation of DOM (Gu et al., 2024; Zhang et al., 2022; Fan et al.,
2024), suggesting a substantial incorporation of O-containing functional groups into
carbon structures during the photolysis and OH oxidation. The $O/C_w$ and $OS_{c,w}$
values for YL WSOM demonstrated analogous changes following OH
photooxidation, increasing from 0.46 to 0.57 and from -0.11 to 0.11, respectively.
However, the $O/C_w$ and $OS_{c,w}$ values for YL WSOM exhibit slight decrease after
photolysis, declining from 0.46 to 0.43 and from -0.11 to -0.15, respectively. These
findings indicate that the photochemical evolution of WSOM is significantly
influenced by their molecular composition. Nonetheless, it is undoubtedly that the
$O/C_w$ and $OS_{c,w}$ values of aged WSOM resulting from OH photooxidation are
significantly higher than those resulting from photolysis, indicating a more robust
oxidation process.
To further elucidate the molecular distribution of WSOM, van Krevelen (VK)
diagrams were constructed by plotting the H/C ratio versus O/C ratio. As indicated in



Figure S1, the identified compounds were classified into seven distinct regions (Sun et
al., 2023; Song et al., 2018): (I) lipids-like, (II) protein/animo sugars, (III)
carbohydrates-like, (IV) unsaturated hydrocarbons, (V) lignin/CRAMs-like, (VI)
condensated aromatic, and (VII) tannins. It is obvious that lignin/CRAMs-like
compounds emerged as the predominant constituents, comprising 83.1% and 88.4% of
the fresh RS and YL WSOM (Table S2), respectively. The proportion of these
compounds remained stable following photolysis; however, a decline was observed
following OH photooxidation, with the contents decreasing from 83.1% to 63.3% for
RS WSOM and from 88.4% to 73.9% for YL WSOM. Lipid compounds were also
identified in both fresh RS and YL WSOM, with relative higher contents in RS
WSOM, however, a significant reduction in lipid content was noted after OH
photooxidation for both RS WSOM and YL WSOM. This trend aligns with
observations of DOM under UV irradiation, where lignin and lipids were identified as
the most active component involved in molecular conversion (Gu et al., 2024).
Conversely, the content of tannins-like substances in both RS and YL WSOM greatly
increased due to OH photooxidation. This suggest that the attack by OH radical lead
to the formation of more polar tannins compounds, indicating the potential
contribution of multiple oxygen-enrich groups (i.e., carboxyl) to the aged WSOM.
Such additional functional groups may enhance the polarity and reactivity of WSOM,
thereby influencing their optical properties, chemical reactivity, and interactions with
other atmospheric components. It is noteworthy that more condensated aromatic
compounds were observed in aged WSOM subjected to photochemical process,





especially OH photooxidation (e.g., the left and bottom of VK diagrams, Figure S1).
Furthermore, as shown in Table S2, the content of condensated aromatic compounds
increased from 1.08% to 1.55% and 4.86% (RS WSOM) and 2.86% to 4.08% and
5.38% (YL WSOM) after photolysis and OH photooxidation, respectively. These
findings strongly support the notion that condensated aromatic molecules are formed
through the photochemical reactions, particularly the OH photooxidation reaction.
**3.3.2. Comparison of the transformation of WSOM induced by photolysis**
**and OH photooxidation**

To enhance the understanding of molecular transformations occurring in RS and

YL WSOM, the photochemical resistant, degraded, and produced molecules were
investigated. The formulae identified both before and after photochemical aging were
assigned to resistant; unique formulae before reaction represented the degraded
molecules; whereas unique formulae after reaction were considered to newly
produced molecules (Figure 4) (Gu et al., 2024; Fan et al., 2024; Zhao et al., 2022). It
is important to acknowledge that the molecules categorized as resistant may also
include those generated from the photochemical reaction, but with the formulae to
those in fresh molecules. As presented in Table S3, approximately 14.9% of the total
number of formulas in fresh RS WSOM and 23.1% in YL WSOM were degraded
through photolysis, resulting in the formation of 26.0% (RS WSOM) and 31.7 % (YL
WSOM) of newly produced formulae in the aged WSOM, respectively. In contrast,
much higher content of formulae (61.6% of RS WSOM and 65.0% of YL WSOM)
were degraded by OH photooxidation and led to higher content (57.0%-61.0%) of



new formulae. These findings suggest that OH photooxidation possesses greater
oxidative potential, resulting in a more substantial degradation and transformation of
molecules.
As shown in Figure 4, the majorities of the degraded molecules were found in
regions characterized by in O/C (<0.6) within VK diagram. In contrast, the newly
produced molecules were concentrated in the regions of 0.3 < O/C < 0.9. This finding
implies that molecules with low O/C underwent oxidation during photolysis and OH
photooxidation processes, resulting in their transformation into oxygen-enriched
structures, especially through OH photooxidation. Furthermore, notable differences
were observed in the VK diagrams corresponding to photolysis and OH
photooxidation. It is obvious that the degraded molecules from RS and YL WSOM
were distributed in the same region of the VK diagrams; however, the newly formed
molecules resulting from distinct photochemical reactions were distributed across
different regions (Figure S2). For example, the molecules produced from the
photolysis of RS WSOM primarily located in the range of 0.3 < O/C < 0.7 and 0.5 <
H/C < 1.7, whereas those generated through OH photooxidation were found in two
separate regions. The majorities of these molecules were concentrated in the range 0.4
< O/C < 0.9 and 0.4 < H/C < 2.0, indicating a higher formation of oxygenated
compounds through OH photooxidation. As illustrated in Figure 4, the presence of
tannin-like compounds in the molecules produced after OH photooxidation were
much higher than that formed after photolysis for both RS and YL WSOM. These
results indicate that the OH photooxidation process substantially enhanced the



abundance of O-containing functional groups within the molecules, as well as the
overall oxidation state of WSOM. Additionally, certain condensated aromatic
molecules were identified in the regions VI in Figure 4 and S2, showing the newly
production of condensated aromatics during OH photooxidation. According to Table
S4, these newly formed condensated molecules were identified in both RS and YL
WSOM, accounting for 11.6% and 4.7% of the total produced molecules (intensity
weighted). These compounds exhibited lower $H/C_w$ (0.55 and 0.60) and $O/C_w$ (0.14
and 0.17) ratios alongside higher $AI_{mod,w}$ (0.76 and 0.77) values, indicating a
predominance of highly aromatic structures. Moreover, they consisted with CHO,
CHON, CHOS, and CHONS, among which CHON is the highest components (57.7%
and 58.0%) for both RS and YL WSOM. It is noteworthy the $O/N_w$ ratios for CHON
and the $O/S_w$ ratios for CHOS were relatively low, suggesting that the N-containing
and S-containing functional group here may mainly comprised with reduced groups.
According to previous studies, the condensated aromatic compounds are usually
assigned to combustion derived BC molecules, i.e., dissolved black carbon. However,
our study suggesting the OH photochemical oxidation may also lead to the formation
of BC-like molecules.
As shown in Table S5, the degraded molecules exhibited lower $AI_{mod,w}$ values
and $O/C_w$ ratio, and higher $MW_w$, $DBE_w$, and $H/C_w$ values compared to the resistant
molecules. For example, the $AI_{mod,w}$ value and $O/C_w$ ratio of the degraded molecules
of RS WSOM are 0.33 and 0.33, respectively, which are lower than the corresponding
values of 0.38 and 0.45 for the resistant molecules during photolysis. Conversely, the



$MW_w$, $DBE_w$, and $H/C_w$ ratio for the degraded molecules are 392, 8.3, and 1.29,
significantly higher than that for the resistant molecules. Similar differences were
noted between the resistant and degraded molecules in RS WSOM subjected to  OH
photooxidation, as well as in YL WSOM underwent both photolysis and  OH
photooxidation. These differences indicate that the WSOM susceptible to
photochemical aging are those molecules with higher molecular weight, double bond
intensity, and aliphatic structures but with lower aromaticity and O-containing group.

In comparison, the newly formed molecules within RS and YL WSOM

demonstrate elevated $O/C_w$, $DBE_w$, and $MW_w$ values. For example, the newly formed
molecules for RS WSOM resulting from photolysis possess higher $O/C_w$ (0.48),
$DBE_w$ (12.0), and $MW_w$ (473) than the degraded molecules. These results indicate that
photolysis generates a greater quantity of high molecular weight compounds, which
also contain more oxygenated functional groups, such carbonyl. Furthermore, the
differences between the degraded and resistant molecules during  OH photooxidation
are pronounced than that during the photolysis process, suggesting that a more
extensive aging reaction occurs during  OH photooxidation. However, the variation in
$AI_{mod,w}$ value between the degraded and produced molecules differ across samples. In
the case of RS WSOM, the produced molecules exhibit higher $AI_{mod,w}$ values than the
degraded molecules during photolysis process, yet they are very similar during  OH
photooxidation. Additionally, distinct changes in $AI_{mod,w}$ values were observed for YL
WSOM during photolysis, where the $AI_{mod,w}$ value of newly produced molecules is
lower than that of degraded molecules, and the proportion of aromatic and condensed



aromatic compounds decreased for YL WSOM after photolysis. These discrepancies
may be due to the produced molecules defined represent only a subset of those
generated during the aging process.

**3.4. Comparison of the photochemical evolution of WSOM from biomass**

**burning and coal combustion**

The photochemical evolution of WSOM originating from BB and CC were

comparably investigated in our study. Our results indicate that the absorption spectra
of RS and YL WSOM exhibit a decreasing trend at ranges below 240 nm during
photolysis and OH photooxidation. However, notable differences were observed
between the two types of WSOM. For example, the $\alpha_{365}$ value for RS WSOM
consistently decreased throughout the photolysis and OH photooxidation, whereas
YL WSOM displayed a progressive increase in this value. These differences may be
attributed to the inherent differences in WSOM derived from biomass burning
compared to that from coal combustion (Cao et al., 2021; Song et al., 2018). The
results indicate that WSOM derived from various sources may undergo distinct
changes in absorbance during photochemical aging, potentially leading to varying
impacts on climate change and radiation balance.

Furthermore, the TFI values for both RS and YL WSOM exhibit a gradual

decline during photolysis and OH photooxidation with no significant differences
between the two. However, the variations in fluorophore composition within RS and
YL WSOM were markedly different. For example, three fluorophores in RS WSOM
remained relative stable during photolysis, while the less-oxygenated fluorophores C3





in YL WSOM gradually decreased. These may indicate that the fluorophores C3 in
YL WSOM are more susceptible to photolytic degradation. These results suggest that
the molecular composition of identical fluorescent component in WSOM derived
from different sources may exhibit notable differences.

As previously discussed, there are notable similarities in the molecular

alterations observed in RS and YL WSOM after photolysis aging. Specifically, the
$MW_w$ values for both RS and YL WSOM exhibited an increase after photochemical
processes, suggesting the formation of high MW molecules through the
oligomerization reaction and the resistance of high MW ones during photochemical
aging. Furthermore, the $AI_{mod,w}$ values always decreased, while the O/C ratios
consistently increased for the both aged RS and YL WSOM. These results indicate the
broken of aromatic structures and the formation of O-containing groups within
WSOM as a result of photochemical processes. However, distinct differences in
molecular characteristics between RS and YL WSOM were observed, which may
influence their respective changes due to photochemical reactions. For example,
CHON compound in RS WSOM exhibit minor variation after photochemical
reactions, whereas it greatly decreased in YL WSOM. Notably, RS WSOM
experienced a greater degradation of lipids during photolysis and  OH photooxidation,
leading to the production of carbohydrate or tannin-like substances (Table S2).

**4. Environmental significance**

Biomass and coal combustion releases considerable quantities of WSOM into the



atmosphere, which undergo significant photochemical transformations under light
irradiation, resulting in considerable uncertainty regarding its physical and chemical
characteristics, as well as reactivity in the atmospheric environment. The present
study investigated the optical and molecular evolution in WSOM derived from BB
and CC (i.e., RS and YL WSOM) during aqueous phase photolysis and OH
photooxidation. The findings indicate a marked reduction in light absorption of RS
WSOM at 365 nm during photochemical processes and more pronounces for the OH
photooxidation, indicating a stronger photobleaching. In contrast, a notable
photoenhancement was observed for YL WSOM during the photochemical processes.
These results suggest that the alteration in light absorption of WSOM are closely
linked to the chemical composition of fresh WSOM.
At the molecular level, the degradation of aromatic structures within WSOM was
evident, accompanied by the formations of O-containing polar groups (e.g., carbonyl,
carboxyl groups), as a result of the photochemical reactions, particularly during OH
photooxidation. These results indicates that the oxidation degree and severity of OH
photooxidation is much higher than that of photolysis, leading to the varitions in the
optical properties of WSOM. It is worthing noted that the polymerization occurs in
both photolysis and OH photooxidation, especially in OH photooxidation, as
evidenced by an increase in $MW_w$ and the formation of condensed aromatic
compounds. These condensated aromatic compounds exhibit similarities to the
chemical and molecular structures of combustion derived BC molecules.
Therefore, OH photochemical oxidation may a potential formation mechanism of

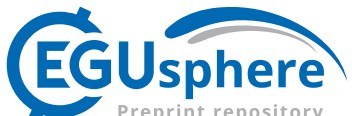

BC-like materials.

It is important to note that this study focused solely on WSOM produced from a

specific type of biomass and coal samples in laboratory simulated system, which may
nor accurately reflect the complexities of combustion processes in real-world
scenarios. Therefore, a more comprehensive investigation into the photochemical
aging of WSOM from diverse biomass and coal sources, as well as under various
conditions in natural environments, is warranted. Furthermore, it is clear that the
photochemical aging processes significantly influences their environmental, climate,
and health effects, necessitating further exploration in future research endeavors.

**Data availability.** The research data can used in this study are available from
Jianzhong Song (songjzh@gig.ac.cn).

**Author contributions.** T. Cao and J. Song designed the research and wrote the paper.
T. Cao, C. Xu, H. Chen and H. Song, analyzed the combustion-derived WSOM
samples during photochemical process. B. Jiang analyzed the WSOM samples by
FT-ICR MS. J. Li, Y. Zhong, and P. Peng commented and revised the paper.

**Competing interests.** The authors declare that they have no conflict of interest

**Acknowledgments.** This study was supported by the National Natural Science
Foundation of China (42192514), Guangdong Major Project of Basic and Applied



Basic Research (2023B0303000007), and the Guangdong Foundation for Program of
Science and Technology Research (2023B1212060049).

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




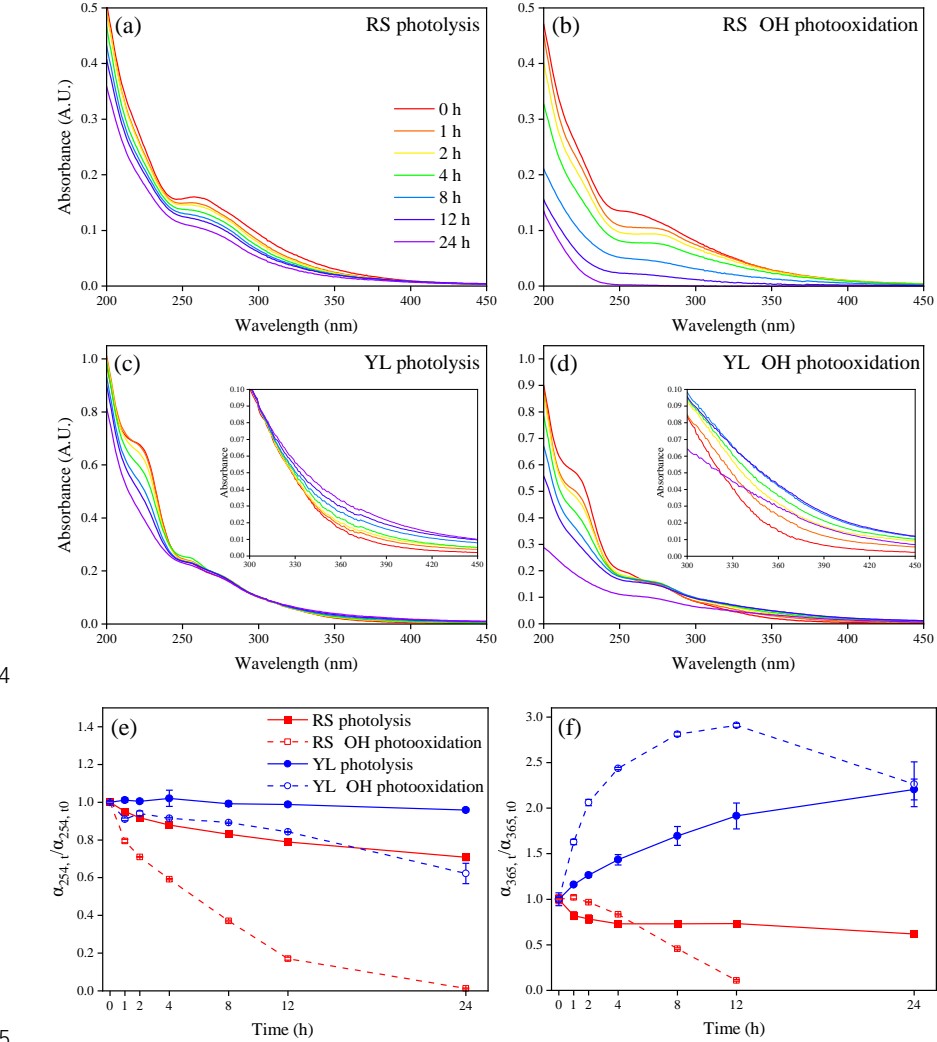



**Figure 1.** UV-vis spectra of RS and YL WSOM during photolysis (a and c) and OH photooxidation (b and d) (The insert figure in c and d represent UV-vis spectra in wavelength range 300-450 nm of YL WSOM during photolysis and OH photooxidation), and changes in $\alpha_{254}$ (e) and $\alpha_{365}$ (f) of RS and YL WSOM during photolysis and OH photooxidation. The error bars represent one standard deviation ($\pm 1\sigma$) of the triplicate samples.







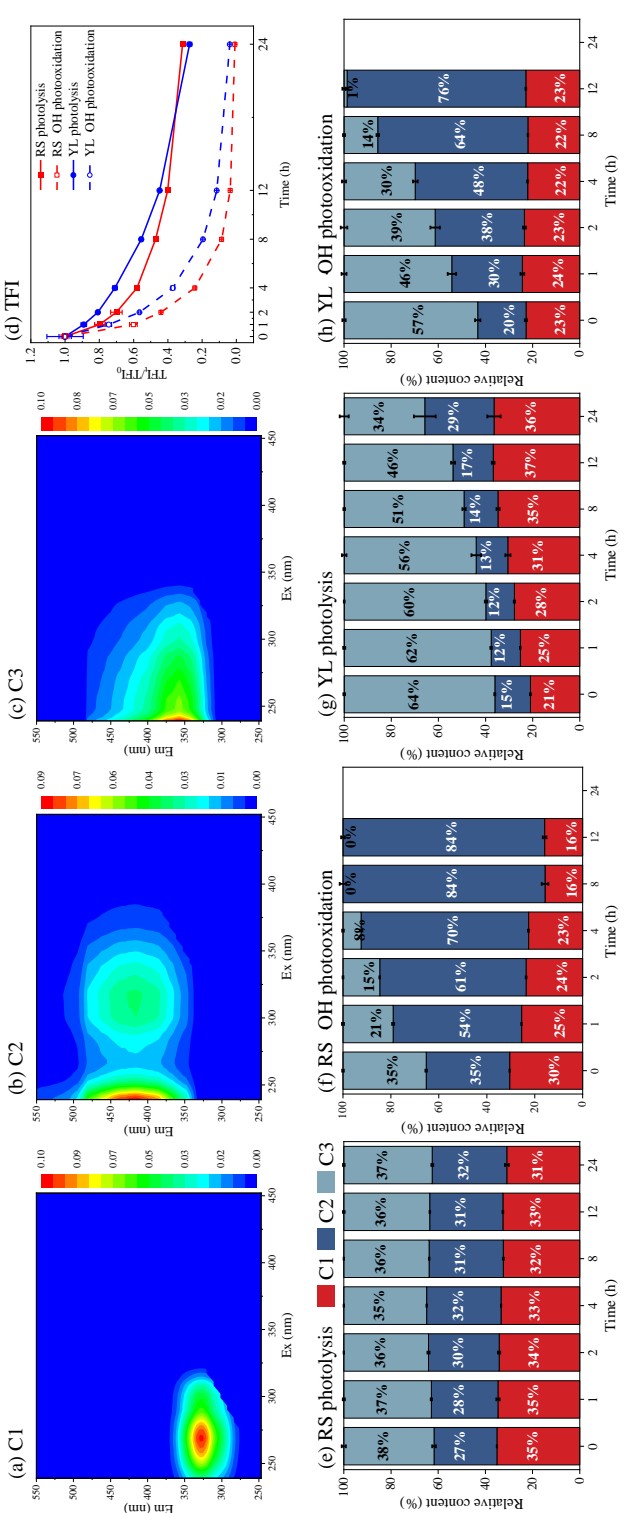


**Figure 2.** (a-c) EEM spectra of PARAFAC-derived fluorescence components (C1–C3) in RS and YL WSOM. Changes of total fluorescence intensity (TFI) (d)
and the relative content of three individual fluorescence component within RS and YL WSOM during photolysis and ·OH photooxidation (e, f, g, h). The
error bars represent one standard deviation (±1σ) of the triplicate samples.



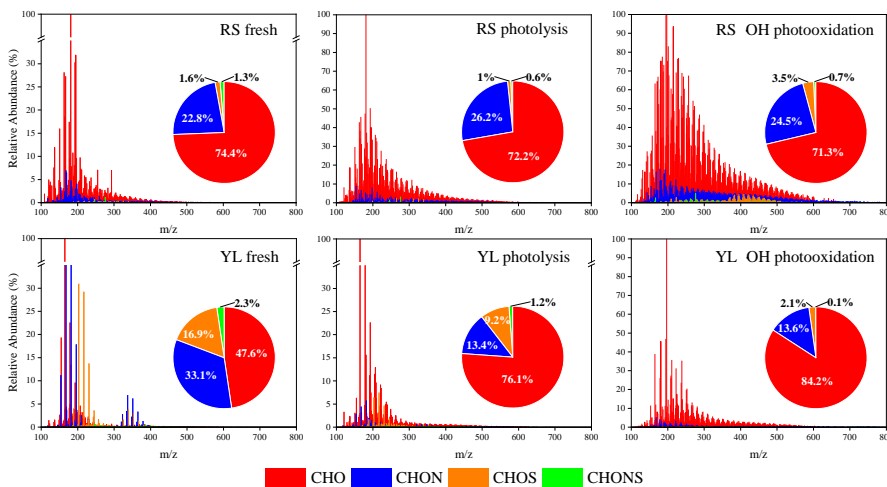

**Figure 3.** Reconstructed mass spectra of RS and YL WSOM for fresh (left), photolysis (middle) and OH photooxidation (right). Pie charts inserted represent the relative content of different formula groups in each sample by sum of intensities of all identified peaks.





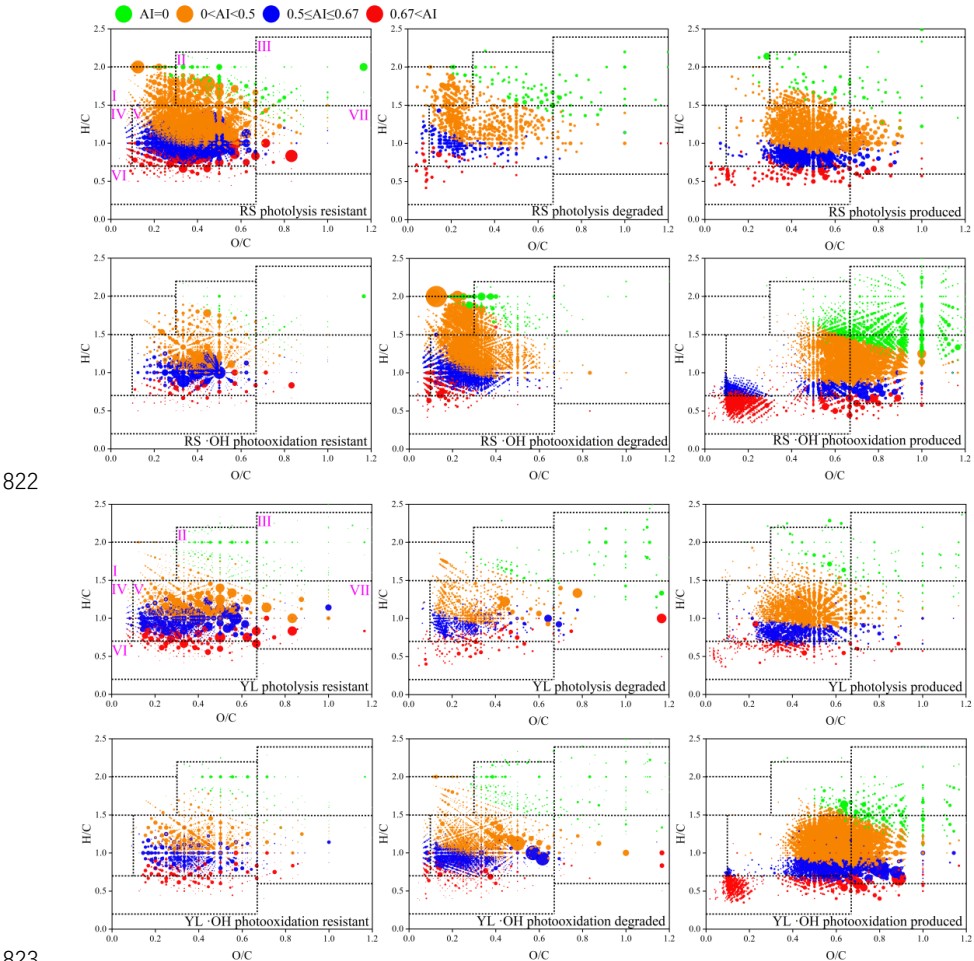

**Figure 4.** Van Krevelen diagrams for molecules resistant, degraded and produced after

photolysis and ·OH photooxidation for RS WSOM (upper) and YL WSOM (bottom).