# Peer review of "Molecular insight into aqueous-phase photolysis and photooxidation"

_EGUsphere, 2025_

## Author Comment (AC1)

**Response to acp-2025-561-RC1**

Comment on acp-2025-561

Anonymous referee #1

This paper investigated the changes in the optical properties, fluorophores, and molecular composition of WSOM derived from the combustion of biomass (RS) and coal (YL) during aqueous photolysis and hydroxyl radical (·OH) photooxidation. Results show distinct photochemical aging effects for RS and YL WSOM, characterized by photobleaching in RS WSOM and photoenhancement in YL WSOM. Additionally, ·OH photooxidation induces more substantial alterations than photolysis, degrading 61.6% of RS and 65.0% of YL WSOM molecules, compared to 14.9% and 23.1% during photolysis, respectively. The oxidation products were characterized by larger molecular weights and higher oxidation levels, including tannin-like substances and a type of black carbon-like compounds, whereas photolysis causes minor changes. These findings are helpful for us to understand the photochemical evolution of combustion-derived WSOM and its environmental and climate impacts. However, I have some questions should be addressed:

Re: We appreciated the reviewer for the valuable comments and suggestions, which are of great help for improving the quality of our manuscript. We have carefully revised the manuscript based on the comments and provided a point-by-point response to all the comments and explained how the comments and suggestions by the reviewer were addressed in the current version of the manuscript.

1. Line 127-128: How about the difference between the light intensity under laboratory conditions (290-400 nm, 12.5 mW/cm$^2$) and the actual solar exposure? How about the actinic flux of the Xe lamp used in present study.

Re: Thanks. In this study, the spectra of Xe lamp and the actual solar exposure were measured using a spectrometer USB2000+ (Ocean Optics, FL, U.S.A.), and the light

intensity of both Xe lamp and solar radiation were measured by an optical power meter (PL-MW2000, Perfect light). As shown Figure S1, the spectrum of Xe lamp is similar with that of the actual solar exposure obtained at noon of May 20, 2023, Guangzhou. The measured light intensity of Xe lamp is 12.5 mW/cm$^2$ (290−400 nm), which is about 5.2 times higher than that of the actual solar exposure.

In addition, the average actinic flux of the Xe lamp was $5.4 \times 10^{-5}$ Einstein/cm²/s¹ (290−400 nm), as determined by the p-nitroanisole/pyridine actinometer method (Laszakovits et al., 2016).

Accordingly, we have added the detailed information in section 2.2 of the present manuscript (Lines 130-135) and Figure S1 in the supporting information (SI).

"The irradiation energy at the water surface is 12.5 mW/cm$^2$ (290−400 nm), which is about 5.2 times higher than the light intensity of actual solar exposure (Figure S1, obtained at noon of May 20, 2023, Guangzhou) and the actinic flux of Xe lamp is $5.4 \times 10^{-5}$ Einstein/cm²/s¹ as determined by the p-nitroanisole/pyridine actinometer method (Laszakovits et al., 2016)." (Lines 130-135)

[Figure]

Figure S1. Spectra of Xe lamp and actual Sunlight (obtained at noon of May 20, 2023, Guangzhou)

Reference:

Laszakovits, J. R., Berg, S. M., Anderson, B. G., O'Brien, J. E., Wammer, K. H., and
Sharpless, C. M.: p-Nitroanisole/Pyridine and p-Nitroacetophenone/Pyridine
Actinometers Revisited: Quantum Yield in Comparison to Ferrioxalate,
Environmental Science & Technology Letters, 4, 11-14,
10.1021/acs.estlett.6b00422, 2016.

2.    For the photolysis and ·OH photooxidation, how many experiments were repeated
in the present study?

Re: Thanks. In the present study, the photolysis and ·OH photooxidation
experiments were both repeated for three times. We have clarified it in the revised
manuscript (Line 125).

3.    Authors are strongly suggested to use the more detailed index: nominal oxidation
state of carbon (NOSC) (Geochimica et Cosmochimica Acta Volume 75, Issue 8,
15 April 2011, Pages 2030-2042), rather than the simplified index: state of
carbon oxidation.

Re: Thanks. According to your suggestion, we have calculated the value of nominal
oxidation state of carbon (NOSC) according to the following formula (LaRowe et al.,
2011) and used NOSC to replace the simplified index (oxidation state of carbon) in the
present manuscript.

$$NOSC = 4 - \frac{4c + h - 3n - 2o - 2s}{c}$$

Here, c, h, n, o, s represents the number of C, H, N, O, S in the formula.

The detailed revisions please refer to Lines 308-324 in main text, and Text S4,
Table S2, S5, and S6 in SI.

References:

LaRowe, D.E. and Van Cappellen, P.: Degradation of natural organic matter: A
thermodynamic analysis. Geochimica et Cosmochimica Acta, 75, 2030-2042,

10.1016/j.gca.2011.01.020, 2011.

4. Line 202: present -> presented

Re: Thanks. We have revised that in the present manuscript. (Line 205)

5. Line 305: The logical subject of the 'with' structure is unclear, it is recommended to replace it with 'while'.

Re: Thanks. We have revised it with "while" in the present manuscript. (Line 307)

6. Line 330: Confirm the spelling of professional terms (such as "condensed" vs. "condensated")

Re: Thanks. We are sorry for this wrong spelling. It should be corrected to "condensed". Accordingly, we have carefully checked the full text and corrected those wrong spelling to "condensed" in the present manuscript. (Lines 330, 347, 349, 352, 393-396, 404, 504).

7. Line 407: What are the similarities between the newly produced condensated aromatic compounds which assigned as BC-like substances and the traditional DBC?

Re: Thanks. It is a good question. According to previous studies, traditional DBC is defined as the water-soluble fraction of black carbon that originates from the incomplete combustion of biomass and fossil fuels (Coppola et al., 2022). The molecular composition of DBC is characterized by high proportion of condensed aromatic compounds and various oxygen-containing functional groups, including carbonyl, carboxyl, and phenolic hydroxyl groups (Liu et al., 2022; Li et al., 2024; Tian et al., 2022). For example, FT-ICR MS analysis of DBC revealed that they are dominated by conjugated aromatic structures and always characterized with low $H/C_w$ ratio (0.15-0.72) and high aromaticity index ($AI_{mod,w}$) values (0.63-0.76) (Fan et al., 2023; Liu et al., 2022; Yan et al., 2022; Tian et al., 2022). In this study, the newly produced condensed aromatic compounds are also displayed with low $H/C_w$ (0.60 and 0.56 for

RS and YL fresh WSOM) and higher $AI_{mod,w}$ values (0.76 and 0.77 for RS and YL WSOM). Obviously, the newly produced condensed aromatic substances (i.e., BC-like substances) and traditional DBC have many similarities in molecular characteristics, such as lower $H/C_w$ ratios and higher $AI_{mod,w}$ values and all comprised of extensive conjugated aromatic structures. Accordingly, we have clarified that in the present manuscript. The detailed revision please refer to Lines 404-409.

"According to previous studies, the condensed aromatic compounds are usually assigned to traditional dissolved black carbon (BC) molecules derived from combustion (Fan et al., 2023; Liu et al., 2022; Yan et al., 2022). These compounds share similar molecular characteristics, such as lower $H/C_w$ ratios and higher $AI_{mod,w}$ values. However, our study suggesting that ·OH photochemical oxidation may also contribute to the formation of BC-like molecules." (Lines 404-409)

References:

Coppola, A. I., Wagner, S., Lennartz, S. T., Seidel, M., Ward, N. D., Dittmar, T., Santín, C., and Jones, M. W.: The black carbon cycle and its role in the Earth system, Nature Reviews Earth & Environment, 3, 516-532, 10.1038/s43017-022-00316-6, 2022.

Fan, J., Duan, T., Zou, L., and Sun, J.: Characteristics of dissolved organic matter composition in biochar: Effects of feedstocks and pyrolysis temperatures, Environmental Science and Pollution Research, 30, 85139-85153, 10.1007/s11356-023-28431-x, 2023.

Li, L., Cheng, W., Xie, X., Zhao, R., Wang, Y., and Wang, Z.: Photo-Reactivity of dissolved black carbon unveiled by combination of optical spectroscopy and FT-ICR MS analysis: Effects of pyrolysis temperature, Water research, 251, 10.1016/j.watres.2024.121138, 2024.

Liu, Y., Wang, M., Yin, S., Xie, L., Qu, X., Fu, H., Shi, Q., Zhou, F., Xu, F., Tao, S., and Zhu, D.: Comparing Photoactivities of Dissolved Organic Matter Released from Rice Straw-Pyrolyzed Biochar and Composted Rice Straw, Environmental science & technology, 56, 2803-2815, 10.1021/acs.est.1c08061, 2022.

Tian, Y. X., Guo, X., Ma, J., Liu, Q. Y., Li, S. J., Wu, Y. H., Zhao, W. H., Ma, S. Y.,
Chen, H. Y., and Guo, F.: Characterization of biochar-derived organic matter
extracted with solvents of differing polarity via ultrahigh-resolution mass
spectrometry, Chemosphere, 307, 10.1016/j.chemosphere.2022.135785, 2022.

Yan, W., Chen, Y., Han, L., Sun, K., Song, F., Yang, Y., and Sun, H.: Pyrogenic dissolved
organic matter produced at higher temperature is more photoactive: Insight into
molecular changes and reactive oxygen species generation, Journal of hazardous
materials, 425, 10.1016/j.jhazmat.2021.127817, 2022.

8. The ·OH oxidation of WSOM in atmosphere can happen in light and dark
conditions. Could your add a discussion between the ·OH photooxidation and dark ·OH
oxidation in the paper?

Re: Thanks for your kindly suggestion. In real atmospheric environment, the ·OH
oxidation of WSOM can occur under both light and dark conditions. Compared with
the results reported in previous studies, some similarities and differences can be
observed between the ·OH photooxidation and dark ·OH oxidation. At first, a
significant reduction in the light absorbance of WSOM, indicative of photobleaching,
was observed during the ·OH oxidation in both light and dark conditions (Fan et al.,
2019). In addition, similar changes at the molecular level were also identified by the
ultrahigh resolution mass spectrometry, such as an increase in the average O/C ratio and
oxidation state of carbon in WSOM, and a decrease in the aromaticity index (AI) values
during both oxidation processes (Fan et al., 2024). These observations suggest that
similar oxidation reactions may happened during the ·OH oxidation of WSOM in both
light and dark conditions. However, distinct differences were also noted between
the ·OH photooxidation and dark ·OH oxidation. For example, the continued
photooxidation by ·OH resulted in the formation of conjugated aromatic compounds,
which share molecular characteristics with traditional DBC and are thus classified as
BC-like substances. In contrast, the dark ·OH oxidation, which is initiated through
Fenton chemistry, did not yield these condensed BC-like compounds (Fan et al., 2024).
This finding implies that the formation of highly aromatic molecules may require

both ·OH oxidation and photoreactions to occur. Accordingly, we have added the detailed discussion in the revised manuscript. (Please refer to Lines 193-195, Lines 266-269, Lines 289-293, Lines 304-305, Lines 312-316, Lines 409-413).

The details are as follows:

"These results are consistent with earlier studies on dark ·OH oxidation of BB WSOM, indicating that the presence of ·OH radicals accelerate the degradation of aromatic structures within WSOM (Fan et al., 2024; Ye et al., 2020)." (Lines 193-195)

"As shown in Figure 3, CHO and CHON compounds are the dominant compounds (95.8%-98.4%) in RS WSOM, with minor fluctuations following photolysis and ·OH photooxidation. Similar changes were also observed for BB WSOM under dark ·OH oxidation (Fan et al., 2024)." (Lines 266-269)

"These observations in molecular weight aligned with findings related to the aqueous-phase photochemical oxidation of wood smoke WSOM, but no significant change in molecular weight was observed in the aqueous-phase dark ·OH oxidation of BB WSOM (Fan et al., 2024; Wong et al., 2019)." (Lines 289-293)

"This significant decreasing in $AI_{mod,w}$ values of BB WSOM was also observed during the dark ·OH oxidation (Fan et al., 2024)." (Lines 304-305)

"These observations are consistent with findings related to BB WSOM under dark ·OH oxidation and the photochemical transformation of DOM (Gu et al., 2024; Zhang et al., 2022; Fan et al., 2024), suggesting a substantial incorporation of O-containing functional groups into carbon structures during the photolysis and ·OH oxidation." (Lines 312-316)

"In contrast, the ·OH oxidation that without the presence of light, initiated by Fenton chemistry, does not produce these BC-like substances (Fan et al., 2024). This possibly indicates that the formation of highly aromatic BC-like molecules requires both ·OH oxidation and photoreactions to occur." (Lines 409-413)

References:

Fan, X., Xie, S., Yu, X., Cheng, A., Chen, D., Ji, W., Liu, X., Song, J., and Peng, P.: Molecular-level transformations of biomass burning-derived water-soluble

organic carbon during dark aqueous OH oxidation: Insights from absorption, fluorescence, high-performance size exclusion chromatography and high-resolution mass spectrometry analysis, Science of the total environment, 912, 169290, 10.1016/j.scitotenv.2023.169290, 2024.

Fan, X., Yu, X., Wang, Y., Xiao, X., Li, F., Xie, Y., Wei, S., Song, J., and Peng, P. a.: The aging behaviors of chromophoric biomass burning brown carbon during dark aqueous hydroxyl radical oxidation processes in laboratory studies, Atmospheric Environment, 205, 9-18, 10.1016/j.atmosenv.2019.02.039, 2019.

9. Fig 3: the name of vertical axis should revised to "Relative intensity".

Re: Thanks. According to the suggestion, we have revised the name of vertical axis to "Relative intensity" in the revised Figure 3.

[Figure]

Figure 3. Reconstructed mass spectra of RS and YL WSOM for fresh (left), photolysis (middle) and ·OH photooxidation (right). Pie charts inserted represent the relative content of different formula groups in each sample by sum of intensities of all identified peaks.

---

## Author Comment (AC2)

**Response to acp-2025-561-RC2**

Comment on acp-2025-561

Anonymous referee #2

The research component of this manuscript examines the changes in optical properties, fluorophores, and molecular composition of WSOM from biomass and coal combustion during aqueous photolysis and hydroxyl radical (·OH) photo-oxidation. The results show that the ·OH photo-oxidation process leads to the degradation of more WSOM molecules, producing products with higher molecular weights and higher levels of oxidation, including tannins and newly formed black carbon analogs. While the photolysis products show relatively little change. These findings provide new insights into the photochemical evolution of WSOM from combustion and help predict its impact on the environment and climate change. In conclusion, this manuscript is highly recommended to be accepted for publication with a few modifications:

Re: Thank you for your valuable criticisms and comments, which are of great help for improving the quality of the manuscript. According to your comments, we have revised the manuscript and provided a point-by-point response to all the comments and explained how the comments and suggestions were addressed in the current version of the manuscript.

1.  Line 102-103: Please provide basic information about the two fuels.

Re: Thanks. We have provided the basic information about the two fuels in the revised manuscript. The detailed information please refer to Lines 101-103 in revised main text, Text S1 and Table S1 in the supporting information (SI).

The details are as follows:

"The detailed information of RS and YL were provided in Text S1 and Table S1 in the supporting information (SI)." (Lines 101-103 in revised main text);

"Rice straw (RS) and Yulin coal (YL) were selected as representative biomass and coal fuel materials for the preparation of combustion-derived WSOM samples. RS was

collected in Chuzhou, Anhui Province. As shown in Table S1, the carbon (C), hydrogen (H), oxygen (O), nitrogen (N) and ash contents of RS were 38.3%, 5.74%, 43.8%, 1.90% and 10.3%, respectively. Prior to combustion, the RS was sorted and cut into smaller sections to enhance the efficiency of the combustion process. YL coal was collected from Yulin, Shanxi Province. The maturity and volatile content of YL coal were determined to be 0.58% and 34.4%, respectively, confirming its classification as high volatile bituminous coal. Additionally, the ash content of YL coal was measured at 4.22% and the C, H, O, N and S contents of YL coal were 77.0%, 6.07%, 11.6%, 1.01% and 0.17%, respectively." (Text S1. Sample information in SI)

Table S1. Basic information and elemental composition of RS and YL (%)

|  | Biomass (RS) | Coal (YL) |
|---|---|---|
| C (%) | 38.3 | 77.0 |
| H (%) | 5.74 | 6.07 |
| O (%) | 43.8 | 11.6 |
| N (%) | 1.90 | 1.01 |
| S (%) | - | 0.17 |
| Ash (%) | 10.3 | 4.22 |
| Maturity (%) | - | 0.58 |
| Volatile (%) | - | 34.4 |
| Rank | - | High volatile bituminous coal |

2. Line 116-119: Please present the WSOC measurement protocol in the manuscript or SI: instrumentation, general experimental methods.

Re: Thanks. According to your suggestion, the WSOM measurement protocol have been added in SI. The detailed information please refer to Lines 116-122 in the manuscript and Text S2 in the SI.

The detailed revision are as follows:

"The organic carbon concentration of WSOM solution was measured by a total organic carbon analyzer (VCPH analyzer, Shimadzu, Kyoto, Japan) following the non-purgeable organic carbon protocol. After the removal of inorganic carbon, the sample

was oxidized at high temperature (680 °C) to generate $CO_2$ and then determined by non-dispersive infrared detector. Before photochemical reaction, WSOM solution was diluted to 20 mgC/L by ultrapure water. The detailed information can be found in Text S2 of SI." (Lines 116-122 in the revised manuscript)

"The weighted smoke filters were fragmented into small pieces and subsequently placed into a pre-baked 100 mL glass bottle. Then, 60 mL of ultrapure water was added, and the mixture was subjected to sonication at a constant temperature of 25°C for a duration of 30 min. The supernatant was filtered through a 0.22 μm PTFE membrane, resulting in a solution that serves as the stock solution of water-soluble organic matter (WSOM). The concentration of the WSOM stock solution was quantified using a total organic carbon analyzer (VCPH analyzer, Shimadzu, Kyoto, Japan) in accordance with the non-purgeable organic carbon protocol. After the removal of inorganic carbon, the sample underwent oxidation at a high temperature of 680°C, and the peak area of $CO_2$ was measured using a non-dispersive infrared detector. The WSOM stock solution was subsequently diluted to approximately 20 mgC/L with ultrapure water, based on the measured concentration. Photolysis experiments will be conducted utilizing this diluted solution." (Text S2 in the SI)

3.    Line 129-132: What is the approximate OH concentration during the corresponding oxidation time? How much $H_2O_2$ is consumed in this process? Is there any estimate?

Re: Thanks for these good questions. In the ·OH photooxidation experiment, WSOM was subjected to continuous reaction with ·OH generated from hydrogen peroxide ($H_2O_2$) under light exposure. The concentration of ·OH in the reaction solution was determined using benzoic acid as a chemical probe, as described by Tong et al. (2015) and Hems et al. (2018). The results shown that the ·OH concentration in the reaction solution varied between $9.3 \times 10^{-14}$ to $1.3 \times 10^{-13}$ mol/L.

In the present study, the initial concentration of $H_2O_2$ in the reaction solution was set at 3 mM, which falls within the range of 0.1 mM to 10 mM reported in prior studies (Fan et al., 2023; Zhao et al., 2015; Ye et al., 2019). In addition, it was observed that about 35.6%±4.6% of $H_2O_2$ was consumed after a 24h reaction period in the present

study, as measured by a N,N-diethyl-p-phenylenediamine (DPD) chemical probe method. This result suggests that there was an adequate supply of ·OH available to react with WSOM in the ·OH photooxidation experiment.

Accordingly, we also clarified that in the revise manuscript (Lines 137-140).

"Based on the conditional experiment, it was found that approximately 35.6%±4.6% of $H_2O_2$ was consumed after a 24-hour reaction period. This indicates that there was a sufficient amount of ·OH present to react with WSOM during the ·OH photooxidation experiment." (Lines 137-140)

Reference:

Fan, X., Xie, S., Yu, X., Cheng, A., Chen, D., Ji, W., Liu, X., Song, J., and Peng, P.: Molecular-level transformations of biomass burning-derived water-soluble organic carbon during dark aqueous OH oxidation: Insights from absorption, fluorescence, high-performance size exclusion chromatography and high-resolution mass spectrometry analysis, The Science of the total environment, 912, 169290, 10.1016/j.scitotenv.2023.169290, 2024.

Hems, R. F. and Abbatt, J. P. D.: Aqueous Phase Photo-oxidation of Brown Carbon Nitrophenols: Reaction Kinetics, Mechanism, and Evolution of Light Absorption, Acs Earth and Space Chemistry, 2, 225-234, 10.1021/acsearthspacechem.7b00123, 2018.

Tong, M., Yuan, S., Ma, S., Jin, M., Liu, D., Cheng, D., Liu, X., Gan, Y., and Wang, Y.: Production of Abundant Hydroxyl Radicals from Oxygenation of Subsurface Sediments, Environmental science & technology, 50, 214-221, 10.1021/acs.est.5b04323, 2015.

Ye, Z., Qu, Z., Ma, S., Luo, S., Chen, Y., Chen, H., Chen, Y., Zhao, Z., Chen, M., and Ge, X.: A comprehensive investigation of aqueous-phase photochemical oxidation of 4-ethylphenol, Science of The Total Environment, 685, 976-985, 10.1016/j.scitotenv.2019.06.276, 2019.

Zhao, R., Lee, A. K. Y., Huang, L., Li, X., Yang, F., and Abbatt, J. P. D.: Photochemical processing of aqueous atmospheric brown carbon, Atmospheric Chemistry and

Physics, 15, 6087-6100, 10.5194/acp-15-6087-2015, 2015.

4. Line 184-185: What physical properties of WSOC can be characterized by α254 and α365?

Re: Thanks. In this study, $\alpha_{254}$ and $\alpha_{365}$ refer to the absorbance coefficients of organic substances at 254 and 365 nm (Fan et al., 2019; Zhao et al., 2023). The $\alpha_{254}$ value indicate the content of aromatic structures and double bonds, which are associated with $\pi$ electron clouds that are prone to transition under ultraviolet light excitation at around 254 nm. In general, it was positively correlated with the amounts and properties of aromatic components of organic matter and had been widely applied to indicate the light absorption of WSOM in previous studies (Fan et al., 2019). The $\alpha_{365}$ represent the light absorption of WSOM in near ultraviolet and visible ranges. Previous research has demonstrated a strong correlation between $\alpha_{365}$ and the cumulative absorption measured between 300 and 400 nm, which had been widely to indicate BrC in atmospheric environments (Hecobian et al., 2010; Jiang et al., 2021). Accordingly, we have added a brief description in the revised SI. Please refer to Text S3 in the SI.

"The $\alpha_{254}$ value indicates the content of aromatic structures and double bonds, which was positively correlated with the amounts and properties of aromatic components. The $\alpha_{365}$ value represents the light absorption of WSOM in near ultraviolet and visible ranges, which had been widely used to characterize BrC in atmospheric environments (Hecobian et al., 2010; Jiang et al., 2021b)." (Text S3 in SI)

Reference:

Fan, X., Yu, X., Wang, Y., Xiao, X., Li, F., Xie, Y., Wei, S., Song, J., and Peng, P. a.: The aging behaviors of chromophoric biomass burning brown carbon during dark aqueous hydroxyl radical oxidation processes in laboratory studies, Atmospheric Environment, 205, 9-18, 10.1016/j.atmosenv.2019.02.039, 2019.

Hecobian, A., Zhang, X., Zheng, M., Frank, N., Edgerton, E. S., and Weber, R. J.: Water-Soluble Organic Aerosol material and the light-absorption characteristics of aqueous extracts measured over the Southeastern United States, Atmospheric

Chemistry and Physics, 10, 5965-5977, 10.5194/acp-10-5965-2010, 2010.

Jiang, H., Li, J., Sun, R., Tian, C., Tang, J., Jiang, B., Liao, Y., Chen, C.-E., and Zhang, G.: Molecular Dynamics and Light Absorption Properties of Atmospheric Dissolved Organic Matter, Environmental science & technology, 55, 10268-10279, 10.1021/acs.est.1c01770, 2021.

Zhao, Y., Kumar, A., Wang, K., Lin, J., Yu, Z., Cheng, S., Zhang, S., Yu, Z., and Liu, D.: Spatial heterogeneity of soil dissolved organic matter characteristics in the riparian zone of the Three Gorges Reservoir, Ecohydrology, 17, 10.1002/eco.2570, 2023.

5. Line 350-354: The results in this section show that the content of condensed aromatic compounds increases after photolysis, but the results of α254 and α365 of rice straw calculated in the absorption section show that the aromaticity decreases. Is there any connection or difference between the results in these two sections? Please explain

Re: Thank you for your question. We agreed with your observation that the content of condensed aromatic compounds increases after photolysis, but the results of $\alpha_{254}$ and $\alpha_{365}$ of RS WSOM decreased. These differences can be explained by the fact that these two parameters characterize different fraction of WSOM. The $\alpha_{254}$ and $\alpha_{365}$ of WSOM indicate the light absorption properties of total WSOM. Therefore, the results of $\alpha_{254}$ and $\alpha_{365}$ of RS WSOM calculated in the absorption section showed that the aromaticity of WSOM decreased after photolysis. However, the condensed aromatic compounds only contributed 1.08% and 4.86% of all identified molecules for fresh and ·OH protoxidized RS WSOM, which are too small to represent the total molecular properties of WSOM. The content of condensed aromatic compounds increases after photolysis only indicated the formation of condensed aromatic fraction rather than the aromaticity changes of total WSOM. Therefore, we think these different results for the changes of the content of condensed aromatic compounds and the $\alpha_{254}$ and $\alpha_{365}$ values of RS WSOM after photolysis and ·OH photooxidation are scientifically reasonable.

6. Figure 4: It is recommended that the sub-graphs in Figure 4 be numbered to make

it easier to distinguish the results for coal and biomass.

Re: Thanks. In the revise manuscript, we have numbered the sub-graphs in Figure 4. In addition, we also revised the manuscript based on the revised Figure 4. The detailed revision please refer to Lines 373, 388, 394, and the revised Figure 4.

[Figure]

Figure 4. Van Krevelen diagrams for molecules resistant, degraded and produced after photolysis and ·OH photooxidation for RS WSOM (upper) and YL WSOM (bottom).

---

## Author Response (AR2)

Dear Joachim Curtius,

Thank you for your detailed review of our manuscript. According to your comments, we have carefully read and revised that in the current version of manuscript. In addition, we also revised the the abbreviation WSOM in the "Short summary" based on the suggestion of Katja Gänger (Editorial Support). The detailed revisions are as follow:

1) Many thanks for the detailed revisions answering all comments by the reviewers. Just one technical edit from my side: In line 408 of the revised manuscript, please change "However, our study suggesting that..." to "However, our study suggests that... ".

Re: Thanks. We have revised "However, our study suggesting that..." to "However, our study suggests that... " in the current version of manuscript. (Line 408)

2) Notification to the authors: Your "Short summary" system section includes the abbreviation WSOM. Please adapt your short summary avoiding abbreviations to make it better understandable for non-experts and please pay attention to use only 500 characters including spaces.

Re: Thanks. We have revised the abbreviation WSOM in the "Short summary" system section to " water-soluble organic matter (WSOM)", according to the suggestion.

Please do not hesitate to contact me if you have any questions for the attached files.

Best regards,
Jianzhong Song
* * *
Guangzhou Institute of Geochemistry, Chinese Academy of Sciences
511 Kehua Street
Guangzhou 510649, PR China
Email: songjzh@gig.ac.cn